# Eating for honour: A cultural-ecological analysis of food behaviours among adolescent girls in the southern plains of Nepal

Joanna Morrison[1]*, Machhindra Basnet[2], Neha Sharma[2]

**1** UCL Institute of Global Health, London, United Kingdom, **2** MIRA, Thapathali, Kathmandu, Nepal

* joanna.morrison@ucl.ac.uk

**Data Availability Statement:** Data cannot be shared publicly because participants did not consent to this. Requests for access to the data are reviewed by an institutional review committee

## Abstract

Access to adequate and nutritious food is important for the current and future health of adolescent girls. Interventions often focus on the individual as responsible for their own health ignoring the complex structural issues that underlie optimal nutrition. In South Asia gender inequalities have been noted as an important determinant of poor nutrition among women and their young children, but analysis of adolescent girls' diets and what influences these are rarely undertaken. Therefore, we sought to analyse the factors affecting what and where girls' eat and what affects their behaviour in the plains of Nepal, using a cultural-ecological approach. We analysed a secondary qualitative dataset of focus group discussions with adolescent girls aged 12–19 years old, young mothers, mothers-in-law, and older female key informants. Eating was heavily influenced by patriarchal norms. Boys had preferential access to food, money, and freedom of movement to appreciate their future role in providing for the family. Food was an investment, and boys were perceived to have more nutritional need than girls. Girls were not perceived to be a good return on investment of food, and eating practices sought to prepare them for life as a subservient daughter-in-law and wife. Obedience and sacrifice were valued in girls, and they were expected to eat less and do more housework than boys. Girls' eating and behaviour was constrained to maintain self and family honour. Interventions should acknowledge cultural influences on eating and engage multiple actors in addressing harmful gender norms which limit eating and prevent girls from reaching their potential.

## Introduction

Ensuring girls' access to a sustained and healthy diet has a profound impact on their health, and the health of future generations [1]. Improving nutritional status in adolescence has the potential to address prior nutritional deficits and prevent the development of Non-Communicable Diseases (NCDs) in adulthood. Ensuring girls are well nourished is particularly important in low- and middle-income contexts where girls are often pregnant in adolescence. Pregnancy while malnourished is dangerous for both mother and baby [2]. Maternal

chaired by Dr Saville and Dr Manandhar (n.
saville@ucl.ac.uk).

**Funding:** The original research was funded by the
Department for International Development South
Asian Research Hub Grant number PO 5675, but
no funding was received for this secondary
analysis.

**Competing interests:** The authors have declared
that no competing interests exist.

micronutrient deficiencies affect the development of the placenta and fetus and have negative
impacts on maternal and newborn health outcomes [3]. Adolescent anaemia is common in
LMICs, and adolescent girls have higher iron requirements due to menstruation and growth
spurts [4]. Pregnancy further depletes iron stores. The odds of maternal death are twice as high
in women with severe anaemia than in those without, and studies have shown associations
between anaemia and postpartum haemorrhage [5] and antenatal and postnatal sepsis [6]. If
girls are malnourished, this often results in maternal short stature, which is associated with
low-birth weight and increased newborn mortality.

## Girls' nutritional status in South Asia

Research has shown that in South Asia, population-level factors such as food insecurity, lack of
access to water and sanitation, and lack of access to health care all contribute to poor nutri-
tional status among women and girls [7, 8]. Gender inequalities are also a key and persistent
driver of poor nutrition in this context. A systematic review of literature from South Asia
found that gender discrimination and cultural beliefs about food properties and eating behav-
iours typically caused women to receive comparatively less food than other family members
[9]. A longitudinal study from India found that adolescent boys were more likely to consume
nutritious food than girls, and have higher dietary diversity than girls throughout adolescence.
The gender gap particularly widens at 15 years old [10].

## Factors affecting what and how much girls eat in Nepal

In Nepal, the 2016 Nepal Demographic and Health Survey reported that 17% of women of
reproductive age and 30% of adolescent girls had a Body Mass Index of less than 18.5kg/m$^2$
[11], which is considered a chronic energy deficiency. 44% of adolescent girls in Nepal have
anaemia. More than 52% of girls are married before the age of 18 and 17% of 15–19 year olds
are already mothers or pregnant with their first child [11]. The literature on unmarried adoles-
cent eating has been less well developed than the literature on married women and girls [12,
13], and therefore we describe what is known about married women in this context.

Women often move to their husbands' homes after marriage, where they are in a low status
position and expected to carry out household chores including preparing and cooking food for
the family [14]. Decisions over what is cooked and the amount that is cooked are often con-
strained by men, who buy food for the household, or the mother-in-law who assumes a guard-
ianship role [15]. Daughters-in-law tend to serve food and eat last to maintain ritual purity, to
demonstrate deference, and to ensure that higher status family members have had enough
[15–18]. Eating last has been associated with eating less [19, 20] and having a lower quality
diet. Men are often allocated more food than women [21, 22], based on perceptions of need
and as an acknowledgment of their role in earning to support the family [23]. They also eat
more luxury foods than women [24]. Research has found that physiological differences in
nutritional requirements do not justify these inequalities [25], and women's economic contri-
bution to the household is largely unnoticed [17, 26].

This shows that it is clearly important to understand the broader socio-cultural environ-
ment in order to develop interventions to promote healthy eating that are appropriate to con-
text. Therefore, we draw on the cultural-ecological model [27] to analyse data collected in 2013
to explore what affects girls' eating behaviour [28] in rural plains Nepal. Based on this analysis,
we make recommendations of areas for further research and interventions that may improve
girls nutritional status in this and similar contexts [29, 30].

The cultural-ecological model provides a theoretical structure to organise the determinants
of nutrition and food consumption through five inter-related components of dietary

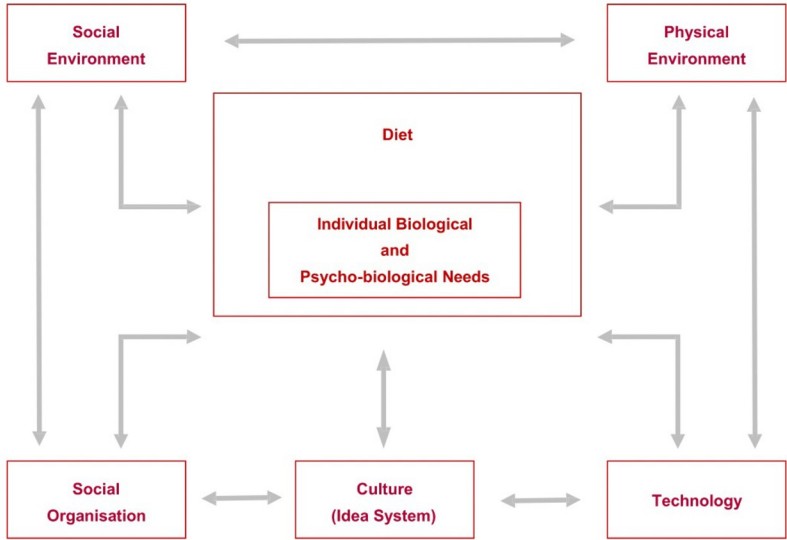

**Fig 1. Cultural-ecological model of food and nutrition.**

behaviors: the physical and social environment where food is acquired; social organisation of how households take care of their members, and how food and eating is organised in the household; technologies related to food production, distribution and acquisition and consumption; and culture which refers to knowledge, beliefs, values, perceptions and emotions that relate to the acquisition, preparation and consumption of food. Diets interact with biological and psycho-biological characteristics of individuals to determine their nutritional status [31] (Fig 1).

## Methods

### Setting

Our research was conducted in adjoining districts of Dhanusha and Mahotarri in Province 2 in the plains of Nepal, which has a combined population of 1.4 million [32]. The plains region has plentiful groundwater and large areas of cultivable land, yet poverty is common, and the area has poor nutrition and education indicators relative to most of the country. For example, at the time of data collection, in Dhanusha 41% of women were underweight and 17% had short stature [33]. Child undernutrition is also high with 41% of children stunted and 32% underweight [34] and only 14% of men and 6% of women had completed their School Leaving Certificate [32]. Female literacy rates were 44% in Dhanusha and 39% in Mahotarri [32]. Dhanusha and Mahottari share an open border with Bihar in India to the south and have a distinct ethnolinguistic Maithili culture. Hindus are the dominant religious group, but these districts also have a significant Muslim population. Muslims are recognised as one of the most marginalised and disadvantaged groups in Nepal [35]. The typical meal in Nepal consists of lentils, rice and vegetables which are eaten twice daily [36]. The consumption of fruit and vegetables is lower than recommended [37, 38] and the consumption of ultra-processed foods is Increasing, particularly in urban areas [39, 40]. Ultra-processed foods are often consumed as snacks between meals and consist of 'junk food' such as 'chat-pat', instant noodles, pani puri, donuts, and biscuits [41–44]. Consumption of these foods has been linked to status as these are prepared and eaten outside the home [40, 45].

## Sampling and data collection

The primary study within which our research is nested sought to understand the factors affecting intrahousehold food allocation and nutrition in pregnancy to inform the development of an intervention to reduce the incidence of low birth weight [17, 46]. This secondary analysis [47] of qualitative data by social scientists engaged in the primary study uses a subset of data from girls and women for a focused analysis examining the socio-cultural factors influencing girls' eating in rural plains Nepal.

**Sampling.** As this is a secondary analysis of data, the sampling strategy was not specifically designed to explore the research questions of this paper. Nonetheless it provides a dataset that enables comparative analysis and development of themes around our topic. We purposively sampled unmarried Maithili-speaking adolescent girls who were 12–14 years old, and 15–19 years old. When we found that participants were all Hindu, we purposively sampled one group of unmarried Muslim girls who were 12–14 years old to explore differences. We sampled first time married mothers under the age of 20 with a child under the age of two, and mothers-in-law from different households. Mothers-in-law were usually guardians of daughters-in-law and had an important role in household food management and allocation [17]. Given the time limitations of data collection, we were unable to explore differences driven by socioeconomic status or caste, and therefore focused on sampling those of low-socioeconomic status. We also sampled local female community health volunteers (FCHVs) to explore community norms. FCHVs were also mothers-in-law themselves.

**Data collection.** We collected data in six villages where residents had low socioeconomic status. FCHVs located those meeting inclusion criteria and participants were then approached in their homes by six local, trained, Maithili speaking female researchers. Researchers took informed written or verbal consent to participate from women, girls, and their guardians. We conducted three focus group discussions (FGDs) with adolescent girls, two FGDs with young mothers, two FGDs with mothers-in-law, and two FGDs with FCHVs (Table 1). There were six to eight participants in each discussion and researchers collected data in quiet community spaces that were convenient to participants. Discussions were conducted using topic guides. Each topic guide contained questions about girls' daily routines and eating habits and what affected the amount and type of food that they ate. We also asked participants to compare girls' and boys' eating habits and nutritional needs.

**Data management and analysis.** Data were recorded and transcribed in Maithili before being translated into English. Initially, we used descriptive content analysis [48]. Data were read by MB and JM who made memos on the translated data and then discussed and identified descriptive codes, and agreed on a coding structure. Given the relatively small size of the dataset and the preferences of the researchers, we did not use qualitative analysis software. JM and MB both coded all the data in English with different coloured pens. Each colour represented a

**Table 1. Data collection.**

| Participants | Religion | Focus Group Discussions |
|---|---|---|
| Young mothers | Hindu | 2 |
| Mothers-in-law | Hindu | 2 |
| FCHV | Hindu | 2 |
| 12–14-year-old girls | Muslim | 1 |
| 12–14-year-old girls | Hindu | 1 |
| 15–19-year-old girls | Hindu | 1 |
| **Total** | | **9** |

different code. We discussed any disagreements in coding. MB and JM wrote a short description of the data under each code for each participant in a table, and used the constant comparative method of analysis [49] to compare data from different participant groups. This enabled us to explore patterns in the data and divergent opinions. MB then wrote a summary of each code in English. JM then used the cultural-ecological model to structure the analysis, grouping codes under the four components of the model. There were no data under the category of 'technologies' because this wasn't a focus of the original research questions. The analysis and interpretation were then discussed with a young unmarried local Maithili researcher, NSh, to check our interpretations of the data.

This study was approved by the Nepal Health Research Council (108/2012) and the UCL Ethical Review Committee (4198/001).

## Results

### Physical environment of eating

The physical environment of eating refers to process of acquiring food and incorporates where food is bought and/or produced and the factors affecting that.

Regardless of the economic status of the household, most participants said that at home, boys were usually fed first: "Schoolboys eat first, then girls because people say it's good when boys eat first. They tell us 'Our son is hungry, serve him food first'" (12–14-year-old Hindu girls). Boys also had the most access to luxury foods, such as meat, sweets, and dairy: "The delicious items are given more to the son and less to the daughter" (FCHVs). In better-off households, girls were also given luxury foods, but sons were still prioritised.

Outside the home, boys ate snacks in the market or snack shop several times a day. This led many to believe that boys ate more than girls. Girls usually ate at home, although some schoolgirls also ate snacks from carts outside their school at break times. In two focus groups, girls expressed annoyance as boys could do as they pleased, eating when, where and whatever they felt like, whereas they could not: "Boys move here and there and eat different kinds of foods at different places. But girls don't wander around. They don't even go outside the school. . .They eat *grams* bought from the nearby *Tela* (cart) in their lunch time" (12–14-year-old Muslim girls).

### Social organisation

The social organisation of eating refers to economic and demographic information, and information about how the household is organised to care for its members.

**Investing in boys.** In general, girls were not given money. This curtailed their ability to snack or eat outside of the home and limited their agency to choose what they wanted to eat. Most participants said that girls were not given money because the family could not afford to give money to both boys and girls, and mothers were expected to give money to boys. Boys often spent this money on food: "Our sons tell us to give them ten or fifty rupees so that they can go to the market and eat there. But from where will we give (money) to our daughters?" (FCHVs).

Mothers met boys' demands for money partly because there was a perception that they needed more food to become strong and be prepared for working life when they would support their parents: "In order to become an adult, boys need to eat and drink properly. Then, they can earn more while girls usually stay inside the home" (FCHVs). Girls said: "Mothers and fathers feed their sons more thinking that they will go abroad and will earn there. That's why they feed them more delicious and tasty foods and love them a lot" (15–19-year-old Hindu girls). To ensure boys ate, they were allowed to eat outside of their homes if they wanted

to. A mother-in-law said: "When the sons don't like to eat the food cooked in their home, their parents provide them money and tell them to eat in the market, but they don't say this to their daughters" (Mothers-in-law).

The fact that sons were expected to care for parents later in life meant that mothers were careful to take care of sons, prioritising their needs and preferences. A group of 12–14-year-old girls told us: "Daughters have to go to their in-laws' house. Sons earn and feed their parents, but daughters do not earn. So, they respect sons more than daughters." Only two participants in two adult focus groups felt that girls also had the capacity to look after their parents when they were older, and therefore should be fed equally to boys. Several participants felt that girls and boys should be given equal access to food but said this was rarely the case.

**Dutiful girls.**   Adults expected girls to accept and understand differential treatment of boys and girls, and be satisfied with home-cooked, simple, food: "Daughters are very brave. Daughters understand the condition of their parents. They understand their parents more than the sons can. Daughters eat rice and lentils more frequently. They eat whatever they are given to eat" (Mothers-in-law). The expected sacrifices that girls made to their family and to their parents–– doing housework, worshipping gods, studying hard–– when boys were free to roam–– was also associated with lower food intake. When girls told a researcher that they ate less food, she asked them why? They replied: "Because we are involved in our work. We are girls and we must work at home" (12–14-year-old Muslim girls).

## Culture

Culture refers to the broad domain of knowledge, beliefs values, perceptions and emotions that affect and relate to the acquisition, preparation, and consumption of food. Adolescent development is also considered under this theme.

**Food and physical development.**   Eating well was linked with rapid physical development (*Sharirik vikas/Chittai Badchan*) among girls and boys. There was an emphasis on nourishing them well so that they would be capable of supporting their family: "People feed their children without eating themselves so that they will soon become developed and capable to earn money and feed their parents" (Mothers-in-law).

Girls' eating was also associated with physical development: "Children grow very fast because of eating. When people see a child, they say that they look older than they actually are. Look at my grand daughter, she is only 14 years old, but people say that she is 17 or 18 years old" (Mothers-in-law). FCHVs and mothers-in-law felt that it was not good for girls to develop too early, or get fat, because then they would need to be married sooner: "If daughters are given the same food as sons they will physically develop too soon, and the parents will have to marry her early" (FCHVs). Mothers-in-law felt that preventing the early maturation and marriage of daughters was another reason for differential food distribution between sons and daughters: "Daughters are not given food to eat like sons because they will become developed very soon and then their marriage has to be done" (Mothers-in-law).

**Expecting constraint among girls.**   One group of FCHVs advised girls not to eat too much because they would get fat (developed), and two groups of Hindu girls said that they were criticised for eating too much, as this was a sign of laziness: "They say, 'How much do you eat? If you eat so much you will get fat.' They yell at us saying, 'You don't work and just eat. You will get fat if you eat more.' They think our body will look huge" (12–14-year-old Hindu girls). Constraint in eating was also emphasised when visiting others: "We are told that we shouldn't eat too much or too little when we go to other people's homes as people will criticise us. They suggest us not to eat a lot, but instead to eat the proper amount of food. Mostly, they tell us to eat less" (15–19-year-old Hindu girls). After marriage girls and women often ate

less in their husband's home and were praised for doing so. Participants said that after a few days of being treated as a guest in her husband's home, new daughters-in-law often ate less because they often felt embarrassed, or they needed to show modesty (*laaj*). They sought to avoid criticism: "People make bad comments if (daughters-in-law) eat more in another persons' home and if they eat less, they are praised. They also feel afraid. That is why they eat less food" (Young mothers).

Whilst eating outside the home was criticised in boys and girls, this was expected of boys and therefore they didn't suffer the harsh judgement that girls did. When girls bought food outside the home, they and their families were judged: "They say: 'Where do these girls get so much money? Every day they buy and eat'" (15–19-year-old Hindu girls). It was expected for boys to be independent and eat outside their home, whereas girls were praised for staying at home, and eating within the home: "Adolescent girls don't go outside. They live within their home and therefore they eat within their home. They are considered very good if they stay within their home" (Young mothers). Data show that if a girl is thought to be habituated to eating outside the home and spending money, then she may not be a good wife. To prevent this, girls said that their parents had warned them: "If a girl gets in the habit of eating food from outside the house, then her father-in-law and mother-in-law will abuse her" (15–19-year-old Hindu girls).

**Good food, good behaviour.**   Food from outside the home was considered to be bad for adolescent health. Snack shop food was thought to be unhealthy, 'stale', and the cause of an upset stomach. Girls said: "My parents say that I should eat home prepared food because outside food does not have any benefit to the body" (12–14-year-old Hindu girls). Eating in a proper manner–at home, on time, eating rice and lentils—was an indication that children came from an honorable household where they were taken care of: "There is a difference in the eating pattern of children in different families. Where the guardian cares for their children, the children are good. They have good behavior. They eat properly and study well. Those children who don't follow their parents have bad manners. Those boys and girls eat whatever they like. They don't eat in a proper manner" (FCHVs). Eating well was related to the development of a sharp mind and the ability to contribute to the community: "(Adolescents) get energy when they eat well. The eating habits of people make a difference in their knowledge, wisdom, and all" (Young mothers). Snack shops were also perceived to be places where boys drank alcohol, smoked cigarettes, consume chewing tobacco (*parag*), and other addictive substances. These were considered bad, dishonourable behaviours which girls should not do. Keeping girls away from eating outside the home was also keeping them away from these behaviours, and men and boys that enacted these bad behaviours.

**Protecting honour.**   Girls were also discouraged and prevented from eating outside the house because their families feared that others would gossip about them. Snack shops were frequented by men more than women, and girls should not be in that male environment where their reputation might be tarnished: "People don't say good things about girls when they eat outside. . .they are not allowed to go to the crossroads to eat. . .People recognise them. . .they consider it to be a bad habit. . .They say,'Mr X's daughter is very bad and she has bad manners as she was eating at the place where boys were also eating.' They even say, 'She was eating together with Mr Y's son'" (FCHVs). Mothers-in-law said: "People will laugh. . .the community will consider it bad and make comments. . .They will say that Mr. X's daughter goes to the market to eat. But they don't say anything about the sons. They can eat wherever they like" (Mothers-in-law). If girls did eat outside their homes, they were thought to be 'neglected' by their families and the community would have little sympathy if something happened to them (Young mothers).

In one focus group, mothers-in-law said that girls who frequently went outside the home became more aware of their surroundings: "Their mind is awakened. They become conscious

when they go somewhere, whereas whoever remains within the home, their mind remains only within the home. As you work, you can develop more and more ideas" (Mothers-in-law). This awareness and outside exposure were feared to result in love marriages, which were perceived as severely detrimental to a girl's honour and her family's. Prevention of love marriage and protection of a girl's reputation motivated families to discourage eating outside the home.

## Discussion

Our data show that eating practices were heavily influenced by patriarchal norms in this context and were symbolic of the differential relationship that sons and daughters have with the household. The way that girls ate defined the kind of person that they were perceived to be, and therefore affected their future and the future of their family. Obedience and sacrifice were valued in girls, and they were expected to eat less and do more work at home. Girls' behaviour was closely scrutinised, constrained, and reflected on their self and family honour, and affected their marriageability.

### Relevance of our findings in the current context

We acknowledge that there are many girls and women who are challenging traditional thinking about gender roles, caste, and religion in Nepal [50, 51] and there are also important legal (e.g. the 2015 constitution) and societal changes (e.g. male migration) that have affected household norms since these data were collected [52]. Recent research indicates that male migration does not improve child nutritional status or care-seeking for illness [53] and there is a need for more research on how this may affect adolescent girls' nutrition. Some research has noted an increase in women's control over household resources when their husbands migrate [54], but this is often dependent on who else is in household and women's position within the household [55]. Patriarchal gender norms around boy preference and household roles are still evident [56, 57], and it should not be presumed that women's perceptions about nutritional need of boys and girls, and the mobility of girls will change with women's increased control over resources in male migrant households. Recent studies from similar settings in Nepal [14, 58, 59] illustrate that gender remains a consistent and significant determinant of what is eaten and how much is eaten in this context, but more research is necessary—from different parts of Nepal and with men and boys—to understand the effects of political and economic changes on adolescent nutrition [60].

### Boy preference

Preferential feeding of boys and giving them preferential access to money and freedom of movement was a way of expressing appreciation for the role that they would play in their parent's ongoing well-being. Food was considered an investment, and boys were perceived to have more nutritional need than girls to prepare them to support their family. Girls were not often perceived to provide good returns on investment of food, even after marriage. In low- and middle-income country contexts, addressing food insecurity, and increasing awareness of nutritional need may improve diets, but as long as boys are perceived to be the caretaker and earner, investment in girls, and changes in eating will be slow to change. Increasing the economic empowerment of women in the plains of Nepal is essential in improving their access to nutritious diets and in increasing their agency.

Women and girls' diets often worsen after marriage [59], and poor nutrition has impacts across the lifecourse [61]. Economic, political and ethnic marginalisation often makes it more difficult for women and families to go against dominant social norms, as they may suffer

further as a result (ie difficulty in getting girls married), and therefore state and collective responsibility for girls and women's adequate nutrition and economic empowerment is necessary.

## The role of men and boys in promoting empowered eating

Globally, the role of men and boys in addressing unequal gender norms around eating is under-researched. Research has tended to focus on the nutrition of younger children and the role of mothers [62]. In our study context, men often have access to financial resources and buy food for the household [17], but often do not perceive themselves to have a broader role in ensuring that family members are well-nourished. For example recent research on prevention of anaemia in pregnancy in a plains area of Nepal reported that gender-normative roles may have prevented men from playing a more active role in improving diets during pregnancy [15]. The role of men and boys in improving girls access to nutritious diets and encouraging healthy eating needs to be explored.

## Surveillance of girls

Previous research has described the family and community surveillance of girls in Nepal to ensure that family honour is protected [63]. In this patriarchal context, marriage is between two families or lineages, rather than two individuals [64], and therefore should only take place between families of equal status. The honour and status of the family is influenced by the perceived purity of adolescent girls' [65], and dowry price at marriage may increase if a girls' honour is in doubt. Anthropological research has argued that one of the purposes of regulating girls mobility and behaviour is to prepare them for their future life within marriage, socialising them to behave in ways to retain honour, maintain ritual purity and act appropriately [66, 67]. Our research found that this extended to eating, encouraging girls to accept that they could not eat what they pleased, they could not routinely access economic resources, and they may have to sacrifice their nutritional need for the family. This would help prepare them for life as a dutiful daughter-in-law and wife who is accustomed to being restricted in what they could eat [68].

A landscaping report funded by USAID, UK Aid, and Australian Aid found that girls as young as 10 years old reported mobility restrictions. As girls get older and experience menarche their mobility is more severely restricted [69, 70]. Research throughout Nepal has shown that the surveillance and moderation of girls' behaviour is enacted through encouraging them to stay close to home [71], discouraging mixing with boys [72–75], and taking specific and 'safe' routes to school [76, 77]. In addition, menstrual taboos often restrict girls movement, and their eating, for example their access to milk and yoghurt is restricted in some households, although practices vary depending on context [70]. Our research shows that constraints on girls behaviour and surveillance affected what they ate and where they ate. Nepal's multisectoral nutrition plan emphasises increasing awareness of the nutritional needs of girls, addressing menstrual food taboos, and integrating nutrition into school health programmes [78]. These are promising strategies but research has shown that increased knowledge does not necessarily translate into changed eating habits [79], and ensuring the engagement of families and communities in these strategies, and addressing the underlying causes of malnutrition, such as gender inequalities, need to be prioritised in such plans.

## Agency and improved nutrition

Constraints on the agency of girls and women affect their health, education and life opportunities [80]. It is important to increase their agency, but this may not result in healthier diets

without addressing the broader cultural-ecological components of how food is produced, distributed, acquired, and consumed [81]. Snacking outside the home may be perceived as an empowering, rebellious, progressive step particularly for girls whose preferences cannot be expressed within the home due to the need to prioritise and defer to men and boys. Food that is not available at home, or non-traditional food, may be considered prestigious and associated with autonomy, modernity and agency [79]. Constraining food choices for adolescent girls may inadvertently increase preferences for certain foods, such as snack foods [82]. There is a need to take an adolescent-centred approach to reforming food systems, making healthy snacks available, affordable, appealing and aspirational [83]. At present snack shops may not be a safe space for girls [84] and may encourage unhealthy eating. Recognising and increasing adolescent agency around healthy food choices while regulating harmful marketing of unhealthy foods is important in improving diets [79]. In Nepal, it will be important to make snacking spaces safe, and ensure girls are equipped with knowledge and access to healthy snack food as their agency increases.

## Limitations

Given that this is a secondary analysis of data there were weaknesses in our sampling. We did not collect data from boys, men, or religious leaders and therefore we were limited to exploring triangulation in the data among girls, FCHVS, young mothers and mothers-in-law. Some research has shown that fathers give adolescent girls money for snack foods, and the motivations and effect of this are important to explore [85]. There is a need to conduct research with boys, whose health was perceived to be more functionally useful, but their unhealthy habits were tolerated and encouraged by mothers and communities.

We did not collect information on caste from participants, nor explicitly discuss the effect of caste on eating. We felt that asking specifically about caste and socioeconomic status would be divisive in focus groups, and instead sought a sample of participants who lived in areas of low socioeconomic status who could discuss community norms. Our sample was small, and we were unable to sample to saturation, but our methods were rigorous, and our findings are well triangulated. The Muslim sample of girls was too small to generalise from, and their experience differed little from that of Hindu girls, perhaps because of the interaction of religion with dominant patriarchal norms, and acculturation of Muslims in this context [86].

Our data were collected in 2013 and so are somewhat dated, however the lack of research about adolescent eating and nutrition in Nepal makes this study an important contribution to the literature. Recent research indicates that the cultural norms discussed in this paper still constrain girls and women [15, 45, 57, 58], and our findings provide an important description of a particular point in time.

## Conclusion

Our research has demonstrated the need to integrate contextual analysis into the planning and implementation of strategies to address adolescent undernutrition and the cultural-ecological model is a useful framework to use to analyse eating behaviours. We recommend more research about eating behaviours in different contexts within Nepal; with different caste and ethnic groups; and with men and boys to understand their eating and how they might contribute to addressing harmful gender norms. We also recommend state, community and family engagement in efforts to increase the economic empowerment of women, and to increase the agency of girls. Our research shows that health education and interventions focusing only on girls as agents of their own health and eating will have limited effectiveness. We recommend an adolescent-centred approach to strategies to improve eating behaviours, ensuring that

healthy food is available, affordable, appealing, and aspirational. Multi-sectoral approaches which acknowledge and address the structural and socio-cultural constraints on families and girls are needed to change eating, and ultimately address a significant driver of ill health in low- and middle-income countries.

## Supporting information

**S1 Questionnaire.**
(DOCX)

## Acknowledgments

We would like to thank the LBWSAT trial team, Dr Peter Graif, Prof Audrey Prost, and all the participants who took part in this study.

## Author Contributions

**Conceptualization:** Joanna Morrison.

**Data curation:** Joanna Morrison, Machhindra Basnet.

**Formal analysis:** Joanna Morrison, Machhindra Basnet, Neha Sharma.

**Methodology:** Joanna Morrison.

**Writing – original draft:** Joanna Morrison.

**Writing – review & editing:** Joanna Morrison, Machhindra Basnet, Neha Sharma.

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
