## [Decision Letter · Decision Letter 0]

26 May 2023

PONE-D-23-02072Eating for honour: a cultural-ecological analysis of food behaviours among adolescent girls in the southern plains of NepalPLOS ONE

Dear Dr. Joanna Morrison,

Thank you for submitting your manuscript to PLOS ONE. After careful consideration, we feel that it has merit but does not fully meet PLOS ONE’s publication criteria as it currently stands. Therefore, we invite you to submit a revised version of the manuscript that addresses the points raised during the review process.

ACADEMIC EDITOR: The authors needs to incorporate the suggestions by reviewers and frame the manuscript keeping cultural aspects and methodological innovations before submission. Major revision is needed for further processing of the article. 

We look forward to receiving your revised manuscript.

Kind regards,

Ranjit Kumar Dehury

Academic Editor

PLOS ONE

Journal Requirements:

2. PLOS’ questionnaire on inclusivity in global research in your revised manuscript. Our policy for research in this area aims to improve transparency in the reporting of research performed outside of researchers’ own country or community. The policy applies to researchers who have travelled to a different country to conduct research, research with Indigenous populations or their lands, and research on cultural artefacts. The questionnaire can also be requested at the journal’s discretion for any other submissions, even if these conditions are not met.  Please find more information on the policy and a link to download a blank copy of the questionnaire here: https://journals.plos.org/plosone/s/best-practices-in-research-reporting. Please upload a completed version of your questionnaire as Supporting Information when you resubmit your manuscript.

The original research was funded by the Department for International Development South Asian Research Hub Grant number PO 5675, but no funding was received for this secondary analysis. 

The authors received no specific funding for this work.

Additional Editor Comments:

The authors need to incorporate the suggestions by reviewers and frame the manuscript keeping cultural aspects and methodological innovations before submission. Major revision is needed for further processing of the article. 

The authors needs to incorporate the suggestions by reviewers and frame the manuscript keeping cultural aspects and methodological innovations before submission. Major revision is needed for further processing of the article.

Reviewers' comments:

Reviewer's Responses to Questions

**Comments to the Author**

1. Is the manuscript technically sound, and do the data support the conclusions?

Reviewer #1: Yes

Reviewer #2: Partly

2. Has the statistical analysis been performed appropriately and rigorously? 

Reviewer #1: Yes

Reviewer #2: I Don't Know

3. Have the authors made all data underlying the findings in their manuscript fully available?

Reviewer #1: Yes

Reviewer #2: No

4. Is the manuscript presented in an intelligible fashion and written in standard English?

Reviewer #1: Yes

Reviewer #2: Yes

5. Review Comments to the Author

Reviewer #1: 1. The paper describes about the plight of women regarding diet on a theoretical context. There is innovation in investigating such an issue which is prevalent in south Asia. How ever, the article can be improved further and communicated in a better manner.

2. The introduction needs more focused on theoretical approach in a cultural setting. The authors may consider health systems studies to elaborate more particularly the role of diet in a household and overall health outcomes.

3. The methodology need justification of sample, if required in a tabular format.

4. The result section is very lengthy, hence, important finding like cultural practices have to be highlighted.

5. The conclusion have to be very specific and actionable points for the health and related departments.

Reviewer #2: Comments to the author

Thank you for the opportunity to review this work which provides useful insight into the factors that influence adolescent girls diet behaviours in Nepal. While the paper presents useful information on socio-cultural factors that should be targeted, I would recommend that the authors revise the aim of the paper to make it clearer/more concise, as well as to cut down and streamline the introduction (see below). I would also recommend that the manuscript be reviewed for language and flow, as the narrative doesn’t always flow and this can compromise the motivation behind the study and there are several typos, e.g., in line 48, ‘need’ should be replaced with ‘needs’

More detailed feedback is provided below.

General

Abstract

- Line 16-19: The opening lines of the abstract make a case for the importance of nutrition for adolescent girls and the influence of structural issues, but the authors could make a stronger link between the influence of socio-cultural and structural factors on adolescent girls dietary behaviours

- I recommend specifying the age range included in the study within the abstract

- Please specify the qualitative methodology used to collect the data; e.g. focus group discussions/ individual interviews/both, and the methods used to analyse the data in the abstract

- Line 21: Please be more specific around what you refer to as ‘eating’ – was this the quality and/or quantity of food available/accessible to them?

Introduction

- Line 38-41: While I agree that the points made around early marriage and pregnancy and the particular needs to ensure optimal nutrition for girls, I don’t think the links between early marriage, adolescent pregnancy and the nutritional status of adolescent girls, as well as the implications for themselves and their infants are clear. I suggest revising this to strengthen the linkages for the reader

- Line 42-43: the point ‘Dietary risks are the second leading contributor to Nepal’s burden of disease comes out of nowhere ate the end of the paragraph without links to the narrative being told. I suggest revising the paragraph for better flow

- Line 53-58: The aim presented for the study isn’t very clearly stated and I would argue that lines 56-58 provide information better suited to the discussion, as it relates more to the findings that the study presents/implications than the specific aim. As stated above, I suggest revising the aim to more clearly/concisely state what was done.

- The introductory section is extremely long and I find myself confused by why the aim of the study is presented, and then followed by additional background information on the links between culture and food/ food and Nepal and then aspects related to gender. I would recommend a complete revision of the introduction to more clearly and concisely present this background information, followed by clear reflection of the aim at the end.

Methods

- The data presented is now 10 years old and it is unclear of how representative these findings are to the current situation in Nepal. Have there been any shifts in cultural/social norms in the past decade and, if so, how would these influence the interpretation of the findings/ their applications to informing interventions. The authors should discuss this in the discussion section – while it is briefly touched on in the limitations, more insight into how the authors feel this reflects current practices is warranted.

- It would be useful to, alongside information on the number of FGDs, provide the number of participants per FGD.

- The authors state that it was a preference not to use qualitative analysis software – it would be useful to understand the motivation behind this/to discuss whether this may have been a limitation in the discussion section. It would be useful to also understand more around the coding of the data – the methods state that coding was done using highlighter pens, but what was the basis of the coding and how was this informed. The way it is currently described, the approach seems a bit unstructured and it is unclear how the analysis was compared between researchers – did both complete the coding for all FGDs and this was compared? Or was coding for each FGD only some by one researcher and then the comparisons made only across groups. Also, how was the coding them brought together – e.g. was thematic analysis used to group themes/sub-themes? I think you need to be clearer on your analysis methods and the justification to provide more insight into the ‘robustness’ of the methodology, as well as more clearly present the data.

Results

- The ‘themes’ presented in the results are quite overlapping (e.g. Physical and social environment of eating and social organisation) and it is not clear what has informed the grouping of data/themes. This also means that sub-themes become a bit repetitive. I would suggest revising the information into more clearly defined themes that, in combination, describe the overall factors influencing adolescent girls diets

Discussion

- While the discussion provides useful discussion of the findings around social/cultural norms in Nepal, the authors could strengthen the links between these norms and the impact of what and where adolescent girls eat and the implications of this.

- Perhaps it would be helpful for the authors to comment on how these findings might inform intervention development (e.g. intervention targets/modalities) and any remaining gaps that require explanation. In addition, it would be useful to understand the role of adolescent boys and men in these contexts and this may be worth greater emphasis.

6. PLOS authors have the option to publish the peer review history of their article (what does this mean?). If published, this will include your full peer review and any attached files.

Reviewer #1: **Yes: **Ranjit Kumar Dehury

Reviewer #2: No

---

## [Author Response · Author response to Decision Letter 0]

26 Jul 2023

We thank the reviewers for their comments and have responded to their concerns below.

Reviewer #1: 

The introduction needs more focused on theoretical approach in a cultural setting. The authors may consider health systems studies to elaborate more particularly the role of diet in a household and overall health outcomes.

We use the theoretical framework of the cultural ecological model. The data were not collected in the setting of the health system, nor refer to the health system. We sought to understand eating in the cultural context of the home and community. We have however added to the introduction section to strengthen the linkages between girls’ health and nutrition.

The methodology need justification of sample, if required in a tabular format.

The paper is based on a secondary analysis of data collected for a previous study. The sampling criteria are discussed on page 7 line 149. 

The result section is very lengthy, hence, important finding like cultural practices have to be highlighted.

We discuss our results in reference to the theoretical framework (Figure 1), which highlight several factors affecting eating behaviours. We have edited the results somewhat to make them more concise.

Reviewer #2: Comments to the author

Thank you for the opportunity to review this work which provides useful insight into the factors that influence adolescent girls diet behaviours in Nepal. While the paper presents useful information on socio-cultural factors that should be targeted, I would recommend that the authors revise the aim of the paper to make it clearer/more concise, as well as to cut down and streamline the introduction (see below). I would also recommend that the manuscript be reviewed for language and flow, as the narrative doesn’t always flow and this can compromise the motivation behind the study and there are several typos, e.g., in line 48, ‘need’ should be replaced with ‘needs’

We have proofread the article and made changes. We have also substantially revised the introduction section.

More detailed feedback is provided below.

General

Abstract

- Line 16-19: The opening lines of the abstract make a case for the importance of nutrition for adolescent girls and the influence of structural issues, but the authors could make a stronger link between the influence of socio-cultural and structural factors on adolescent girls dietary behaviours

We have added to the abstract as suggested.

- I recommend specifying the age range included in the study within the abstract

We have added the age range.

- Please specify the qualitative methodology used to collect the data; e.g. focus group discussions/ individual interviews/both, and the methods used to analyse the data in the abstract

We have added the methods to the abstract and specified that we used the cultural -ecological framework to analyse the data.

- Line 21: Please be more specific around what you refer to as ‘eating’ – was this the quality and/or quantity of food available/accessible to them?

Our article focuses on eating, which refers to food consumption behaviour and all the factors that affect this. We have added this to the abstract to clarify.

Introduction

- Line 38-41: While I agree that the points made around early marriage and pregnancy and the particular needs to ensure optimal nutrition for girls, I don’t think the links between early marriage, adolescent pregnancy and the nutritional status of adolescent girls, as well as the implications for themselves and their infants are clear. I suggest revising this to strengthen the linkages for the reader

- Line 42-43: the point ‘Dietary risks are the second leading contributor to Nepal’s burden of disease comes out of nowhere ate the end of the paragraph without links to the narrative being told. I suggest revising the paragraph for better flow

We have revised this paragraph and added references.

- Line 53-58: The aim presented for the study isn’t very clearly stated and I would argue that lines 56-58 provide information better suited to the discussion, as it relates more to the findings that the study presents/implications than the specific aim. As stated above, I suggest revising the aim to more clearly/concisely state what was done.

We state the study aim on page 3 “to analyse what affects girls’ eating behaviour in rural plains Nepal.” 

We have removed the lines suggested and adjusted the text on page 6 about the study aim.

- The introductory section is extremely long and I find myself confused by why the aim of the study is presented, and then followed by additional background information on the links between culture and food/ food and Nepal and then aspects related to gender. I would recommend a complete revision of the introduction to more clearly and concisely present this background information, followed by clear reflection of the aim at the end.

We have revised and shortened the introduction section, focusing on gender as a determinant of nutrition. We have moved information about the Nepali diet to the context section of the methods because our results about food and snacking needs this contextual information to be understood.

Methods

- The data presented is now 10 years old and it is unclear of how representative these findings are to the current situation in Nepal. Have there been any shifts in cultural/social norms in the past decade and, if so, how would these influence the interpretation of the findings/ their applications to informing interventions. The authors should discuss this in the discussion section – while it is briefly touched on in the limitations, more insight into how the authors feel this reflects current practices is warranted.

Our data represent a point in time, and due to the lack of research with this group in this setting, our findings are important to present. Although we find it difficult to speculate about exactly what the situation is for girls today without supporting evidence from girls or other research with girls, we have added a paragraph discussing the implications of some social changes, and note that gender inequalities persist in the plains areas of Nepal.

We have also added to the introductory section to demonstrate that the issue of gender is a persistent barrier to good nutrition of girls and women and then cited recent references (ie in the past few years), as well as a systematic review which contains some older references. 

- It would be useful to, alongside information on the number of FGDs, provide the number of participants per FGD.

We have added information in the text about how many participants were in the FGDs (line 349).

- The authors state that it was a preference not to use qualitative analysis software – it would be useful to understand the motivation behind this/to discuss whether this may have been a limitation in the discussion section. It would be useful to also understand more around the coding of the data – the methods state that coding was done using highlighter pens, but what was the basis of the coding and how was this informed. The way it is currently described, the approach seems a bit unstructured and it is unclear how the analysis was compared between researchers – did both complete the coding for all FGDs and this was compared? Or was coding for each FGD only some by one researcher and then the comparisons made only across groups. Also, how was the coding them brought together – e.g. was thematic analysis used to group themes/sub-themes? I think you need to be clearer on your analysis methods and the justification to provide more insight into the ‘robustness’ of the methodology, as well as more clearly present the data.

Researchers did not have equal access to software, and we felt it was important to undertake collaborative analysis. Using the software may have prevented this collaborative approach. We have added description of how we did the analysis.

Results

- The ‘themes’ presented in the results are quite overlapping (e.g. Physical and social environment of eating and social organisation) and it is not clear what has informed the grouping of data/themes. This also means that sub-themes become a bit repetitive. I would suggest revising the information into more clearly defined themes that, in combination, describe the overall factors influencing adolescent girls diets

As stated in our methods section, we used the cultural ecological framework to group the data and themes. We have added description of what is considered under each component of the framework at the beginning of each section to remind the reader of the structure.

Discussion

- While the discussion provides useful discussion of the findings around social/cultural norms in Nepal, the authors could strengthen the links between these norms and the impact of what and where adolescent girls eat and the implications of this.

- Perhaps it would be helpful for the authors to comment on how these findings might inform intervention development (e.g. intervention targets/modalities) and any remaining gaps that require explanation. In addition, it would be useful to understand the role of adolescent boys and men in these contexts and this may be worth greater emphasis.

We have substantially edited our discussion section, adding in sections on the relevance of our findings in the current context, and the need to consider men and boys. In our discussion we have made several recommendations which are summarised in our edited conclusion section. 

Response to editors comments:

Please use the following funding statement:

The original research was funded by the Department for International Development South Asian Research Hub Grant number PO 5675, but no funding was received for this secondary analysis.

As stated in our submission, we did not get approval from study participants to share the original data, and therefore there are ethical reasons why we cannot share our data.

We note that editors request:

The authors need to incorporate the suggestions by reviewers and frame the manuscript keeping cultural aspects and methodological innovations before submission. Major revision is needed for further processing of the article. 

We are not sure what ‘keeping cultural aspects and methodological innovations’ refers to. We would welcome advice on this if further changes are required.

---

## [Editor Report · Decision Letter 1]

8 Aug 2023

Eating for honour: a cultural-ecological analysis of food behaviours among adolescent girls in the southern plains of Nepal

PONE-D-23-02072R1

Dear Dr. Joanna Morrison,

We’re pleased to inform you that your manuscript has been judged scientifically suitable for publication and will be formally accepted for publication once it meets all outstanding technical requirements.

Kind regards,

Ranjit Kumar Dehury

Academic Editor

PLOS ONE

Additional Editor Comments (optional):

Dear Authors,

After reviewing the compliance report and my personal reading the article is found to be of publishable quality. Hence, there is requirement of fine tuning and improvement of the readability.

With regards,

Ranjit
---

## [Editor Report · Acceptance letter]

10 Aug 2023

PONE-D-23-02072R1 

Eating for honour: a cultural-ecological analysis of food behaviours among adolescent girls in the southern plains of Nepal 

Dear Dr. Morrison:

I'm pleased to inform you that your manuscript has been deemed suitable for publication in PLOS ONE. Congratulations! Your manuscript is now with our production department. 

Kind regards, 

on behalf of

Dr. Ranjit Kumar Dehury 

Academic Editor

PLOS ONE